# Effects of Heat Stress and Lipopolysaccharides on Gene Expression in Chicken Immune Cells

**DOI:** 10.3390/ani14040532

**Published:** 2024-02-06

**Authors:** Guang Yang, Xinyi Zhou, Shutao Chen, Anfang Liu, Lingbin Liu, Haiwei Wang, Qigui Wang, Xi Lan

**Affiliations:** 1College of Animal Science and Technology, Southwest University, Chongqing 400715, China; yg1551686096@163.com (G.Y.); a524448729@163.com (X.Z.); cst16634254665@163.com (S.C.); anfangliu@126.com (A.L.); liulb515@163.com (L.L.); 2Chongqing Academy of Animal Sciences, Chongqing 402460, China; wahwe@163.com (H.W.); wangqigui@hotmail.com (Q.W.)

**Keywords:** chicken bone marrow dendritic cells, heat stress, LPS, immunosuppression

## Abstract

**Simple Summary:**

Poultry are influenced by environmental stimuli, which can impact their productivity and immune function. Exploring the mechanisms behind these effects is crucial for better animal husbandry. By examining previous research, we identified stress-related genes and used microarray technology to measure gene expression levels and pathway enrichment. Ultimately, we confirmed that certain immune genes are inhibited by environmental stress. These findings provide preliminary explanations for the decreased immune function in poultry under stress and offer a theoretical basis for improving actual production.

**Abstract:**

Prolonged exposure to high temperatures and humidity can trigger heat stress in animals, leading to subsequent immune suppression. Lipopolysaccharides (LPSs) act as upstream regulators closely linked to heat stress, contributing to their immunosuppressive effects. After an initial examination of transcriptome sequencing data from individual samples, 48 genes displaying interactions were found to potentially be associated with heat stress. Subsequently, to delve deeper into this association, we gathered chicken bone marrow dendritic cells (BMDCs). We combined heat stress with lipopolysaccharides and utilized a 48 × 48 Fluidigm IFC quantitative microarray to analyze the patterns of gene changes under various treatment conditions. The results of the study revealed that the combination of heat stress and LPSs in a coinfection led to reduced expressions of *CRHR1*, *MEOX1*, and *MOV10L1*. These differentially expressed genes triggered a pro-inflammatory response within cells via the MAPK and IL-17 signaling pathways. This response, in turn, affected the intensity and duration of inflammation when experiencing synergistic stimulation. Therefore, LPSs exacerbate the immunosuppressive effects of heat stress and prolong cellular adaptation to stress. The combination of heat stress and LPS stimulation induced a cellular inflammatory response through pathways involving cAMP, IL-17, MAPK, and others, consequently leading to decreased expression levels of *CRHR1*, *MEOX1*, and *MOV10L1*.

## 1. Introduction

Poultry are homothermal animals, which means that maintaining a balanced body temperature is crucial for normal bodily functions. In hot and humid regions, elevated summer temperatures and humidity can affect the thermoregulatory mechanisms of animals, resulting in heat stress [1,2]. Unlike mammals, chickens lack sweat glands and primarily rely on drinking water, panting, and peripheral capillary dilation to dispel heat [3]. Prolonged exposure to heat stress triggers the activation of the hypothalamic–pituitary–adrenal axis (HPA), and the HPA stimulates the secretion of corticosterone [4]. Research indicates that excessive corticosterone binds to lymphocytes, affecting enzyme activity within cells, suppressing the production of immune killer cells, and ultimately affecting the normal functioning of the immune system [5]. Moreover, numerous studies have demonstrated that prolonged heat stress leads to immune organ atrophy and hampers growth and developmental functions [6].

Lipopolysaccharides (LPSs) stand as a critical component within bacterial cell walls [7] and are closely linked to the onset of various diseases, often causing inflammatory responses and septic shock [8]. LPSs play a role in promoting the secretion of various inflammatory cytokines, activating antigen-presenting cells, such as dendritic cells and macrophages, and prompting undifferentiated T cells to differentiate into Th1 cells [9]. Moreover, LPSs serve as potent immune activators, wielding remarkable influence within the body’s immune system [10].

The immune response in animals can be activated by heat stress and LPSs, showcasing interactions between the two [11,12]. Studies have indicated that heat stress heightens the sensitivity of an organism to LPSs, resulting in a more pronounced immune response. Transcriptome sequencing following heat stress treatment in chicken immune organs demonstrated the upregulation of oxidation-related enzyme expression and the downregulation of *MHC-11* and *CD40*. Upon LPS treatment, the number of downregulated genes increased, notably from the WNT family [3]. Consequently, an abnormal modulation of the WNT signaling pathway can lead to concentrated immune-mediated autoimmune and inflammatory diseases, as well as associations with various tumors and cancerous conditions [13]. The relationship between heat stress and LPSs is not entirely unidirectional; LPSs can also influence the body’s response to heat stress. Animals treated with LPSs followed by exposure to heat stress displayed heightened inflammation, as evidenced by elevated levels of interleukins and tumor necrosis factor [14]. In summary, the interaction between heat stress and LPSs is complex. Heat stress augments the body’s sensitivity to LPSs [15], affecting the release and metabolism of LPSs, thereby influencing the body’s immune response. Conversely, LPSs increases the body’s sensitivity to heat stress, exacerbating the inflammatory response [16].

Studies have shown that heat stress affects innate and adaptive immunity [17,18]. LPSs effectively trigger the production of pro-inflammatory cytokine responses in macrophages and neutrophils, making them an ideal factor for constructing inflammation models [19]. The initial experiments conducted a pre-transcriptome sequencing analysis using IPA, identifying 48 interacting genes potentially associated with heat stress and LPSs [20]. Consequently, this study focused on these 48 genes, employing Fluidigm’s 48x48 quantitative microarray to examine the gene expression trends at various time points under heat stress and LPS treatment.

## 2. Materials and Methods

### 2.1. Experimental Animals and Cells

All animal husbandry and handling procedures adhered to the Animal Welfare Management Practices of Southwest University (IACUC NO. Approved: IACUC-20221022-17). A total of 12 high-purity lines of White Leghorn chickens were utilized for this experimental study. Bone marrow cells were obtained from embryos.

### 2.2. Cell Collection

Bone marrow cells were collected from 18-day-old *White Leghorn* chick embryos. Under an aseptic environment, the femur and tibia were taken, and the bone marrow was rinsed with phosphate buffer salt solution to collect the bone marrow cells, filtered and centrifuged, and then the number and activity of cells were detected by using a Trypan blue dye rejection counter. We collected about 1 × 10^7^ cell/mL cells by the Trypan blue rejection method and detected about 95% cell activity.

### 2.3. Experimental Design

The experimental treatment groups included the blank normal temperature treatment group, the temperature control group (TN), the LPS normal temperature treatment group (LPS), the high-temperature treatment group (H), and the LPS heat treatment group (LPS + H). The heat treatment group was exposed to conditions of 45 °C and 5% CO_2_, while the temperature control group was maintained at 41.5 °C and 5% CO_2_. The LPS-treated group was provided with a complete medium containing 200 ng/mL of LPSs sourced from Sigma Aldrich, St. Louis, MO, USA. The control group received a normal, complete culture medium. Cells were collected at three specific time points: 2, 4, and 8 h after the initiation of the experimental treatments.

### 2.4. KEGG Enrichment Analysis

For the gene set functional enrichment analysis, we took both gene expression and candidate genes into consideration and used the KEGG REST API to acquire the most recent gene annotations related to KEGG pathways as the background dataset. Subsequently, we mapped the genes and expressions to this background set and utilized the R software package Cluster Profiler (version 3.14.3, R Core Team, Vienna, Austria) for the analysis. Next, we conducted an enrichment and analysis of the results. The minimum gene set was set at 5, while the maximum gene set was limited to 5000, with a significance threshold of *p* < 0.05 and an FDR of <0.25.

### 2.5. Quantitative Detection

We assessed mRNA expression in the LPS-treated, heat-treated, and LPS + heat-treated groups over 2, 4, and 8 h intervals, respectively. Total RNA extraction was conducted using the RNAqueous^®^ Total RNA Isolation Kit from Ambion, Carlsbad, CA, USA, AM1912. Genetic testing was performed using IFC chips (from Standard Bio Tools Inc., Shanghai, China). A list of primers is detailed in Table A1, with samples and primers loaded onto both sides of the IFC chip. The Biomark HD workflow was subsequently used to detect gene expression levels.

### 2.6. Data and Results Analysis

The raw qPCR data underwent quality control and analysis using real-time fluorescence quantitative analysis software (version 4.8.1 from Standard Bio Tools Inc., Shanghai, China). A least-squares analysis of the primary stimulus factors affecting BMDCs was conducted utilizing JMP Pro 10.0.2 software by the SAS Institute, Cary, NC, USA. The factors considered in the analysis included the chicken strains (White Leghorn chicken), experimental treatments (control, heat treatment, LPS, and heat treatment + LPS), and sampling time points (0, 2, 4, and 8 h). Additionally, interactions between heat stress and LPS treatments were fitted to an ANOVA model. Delta Ct values were normalized against the target gene expression by utilizing the geometric mean of three internal reference genes. The fold change in gene expression was assessed using 2^−ΔΔct^.

## 3. Results

### 3.1. Changes in Gene Expression in LPS- and Heat Stress-Treated BMDCs

We used the IFCs to analyze the expression of candidate genes, which revealed the changes in gene expression over time in different treatment groups (Figure 1). Overall, genes such as *AKAP6, ENPP6, GRM2, MC5R, PDE1A, PDE5A*, and others appeared at a high level at the beginning of the treatment and demonstrated a gradual decrease in expression levels within 4 h, followed by a return to normal. Interestingly, there was a significant change in the expression of three genes, including *CRHR1*, which showcased a consistent decrease across all three treatment groups. When compared with the group treated solely with LPSs, the two groups subjected to heat stress exhibited a more pronounced decreasing trend from 0 h to 2 h. This result suggests that the *CRHR1* gene might inhibit the effects of LPSs during the pre-treatment phase. Instead, the *MEOX1* gene exhibited an initial increase within the first 2 h, followed by a subsequent decrease. Meanwhile, the *MOV10L1* gene showed an overall declining trend. Most of the remaining genes did not exhibit remarkable changes in expression.

### 3.2. Gene Expression Trends in LPS and Heat Stress−Treated BMDCs

Upon an overall assessment of Figure 2, all the genes displayed an initial upregulation in expression at 0 h under the heat stress treatment alone and in combination with the LPS treatment. Subsequently, a gradual return to baseline expression levels was observed. Meanwhile, according to the colors on the heat map, it was evident that the heat stress and LPS treatment group (H + LPS) exhibited higher expression than the heat stress treatment group (H) at 0 h (Figure 2). However, the group treated solely with LPSs showed a more fluctuating pattern in gene expression trends. When considering the gene heat map as a whole, higher gene expression was observed in the upper half. The analysis of the changing trends among the 36 genes exhibiting remarkable differences revealed a clear clustering phenomenon among these candidate genes. For instance, genes within the cAMP family, namely, *PDE5A*, *MC5R*, *AKAP6*, *PDE11A*, *CNR1*, *PDE1B*, and *PDE1A*, were tightly clustered together. They exhibited a similar pattern of upregulated expression from 0 to 4 h under different treatments, followed by a return to normal levels. Furthermore, several genes associated with immunity, such as *SMYD2*, *FOSL2*, *TTC28*, *SRGAP3*, *SLC6A4*, and *NLGN3*, also demonstrated a consistent expression trend and were clustered together. Meanwhile, the genes enriched in the cAMP pathway showed the same changes. There was a decrease in the expression of these genes after 2–8 h for all three treatments.

### 3.3. KEGG Enrichment Analysis

Based on the enrichment analysis of 48 candidate genes, a total of 134 channels were enriched, including 18 significantly enriched pathways. The figure below displays the most significant eight paths according to their P values. The different expressions of genes are mainly enriched to the “MAPK Signaling Pathway, Protein Processing in endoplasmic reticulum, Legionellosis, Viral carcinogenesis, Tuberculosis, Epstein-Barr virus infection, Pathways in Cancer, and IL-17 Signaling Pathway” (Figure 3). There are five pathways related to human diseases: one related to organismal systems, one related to genetic information processing, and the others related to environmental information processing.

Among them, seven differential genes were enriched in “Protein processing in endoplasmic reticulum”, and seven genes were enriched in the “Pathway in cancer”. Additionally, a relatively lesser enrichment was observed in the MAPK signaling pathway and the IL-17 signaling pathway. The MAPK signaling pathway serves as an important mechanism generated by diverse stimuli, acting as a crucial molecule that receives signals from membrane receptors and transmits them into the nucleus. This pathway plays a pivotal role in signaling pathways associated with cell proliferation and differentiation, apoptosis, tumorigenesis, and other essential cellular processes. The IL-17 signaling pathway is linked to cellular inflammatory responses.

## 4. Discussion

In this experiment, quantifying the genes of chicken immune cells under LPS treatment alone, heat stress stimulation, and the dual stimulation of LPS + heat stress revealed that heat stress and LPSs induced similar phenomena of immunosuppression. However, the effect of LPSs alone was notably less than that of heat stress. Additionally, under dual stimulation, the LPSs intensified the immunosuppression of the cells. These findings align with published results indicating that simultaneous exposure to heat stress and LPSs increases follicular granulosa cell death and amplifies the organism’s responsiveness, thereby further elevating its likelihood of generating a response [21]. The line graphs of gene expression at different time points reflected that most of the genes showed an upregulation of expression in the pre-treatment period, followed by a return to baseline within 2 to 4 h (Figure 1). Previous reports suggest that LPSs trigger the activation of the HPA axis, leading to increased CRHR1 expression, which, in turn, induces depressive behaviors in mice [22]. This observation might correlate with poultry displaying reduced appetite and inactivity when exposed to heat stress and LPS stimulation. In mice, *CRHR1* antagonists have shown efficacy in alleviating LPS-induced stress behaviors by inhibiting *CRHR1* [23]. The *MEOX1* gene is implicated in various disease developments. Its inhibition reportedly reduces *CXCR4* levels and the expression of stromal-cell-derived factor 1 (*SDF-1α*), known for its anti-inflammatory effects in blood vessels [24]. This finding could elucidate the sustained reduction in *MEOX1* expression in this experiment, indicating potential anti-inflammatory effects. Present research indicates that the inhibition of *MOV10L1* expression is correlated with platelet distribution [25] and hepatitis infection severity [26]. Strong associations have been found between viruses in plasma, hepatitis, and *MOV10L1* expression [27]. In summary, concurrent heat stress and LPS coinfection resulted in decreased expression levels of *CRHR1*, *MEOX1*, and *MOV10L1*.

Chicken bone marrow dendritic cells, recognized as antigen-presenting cells [28], were the focus of this experiment, which analyzed gene expression patterns at successive time points. The investigation revealed a significant elevation in the expression of genes associated with the cAMP pathway under the dual stimulation of heat stress and LPSs. Conversely, compared to the enrichment observed in immune-related genes, the expression of genes within the cAMP pathway was notably lower (Figure 2). This emphasizes that the combined stimulation of heat stress and LPSs exacerbates the effects of immunosuppression on cells. Prior research indicates that the cAMP signaling pathway primarily operates via the activation and inhibition of cyclase, wherein A-kinase anchor proteins (AKAPs) play a pivotal role in signaling [29]. The release of stress hormones contributes to an increase in cAMP signal transduction [30]. This pathway is acknowledged for its role as an inducer of anti-inflammatory responses and serves as a coordinator in crucial steps aimed at addressing inflammation [31]. The cAMP signaling pathway accomplishes its anti-inflammatory effects by activating cytoplasmic kinase complexes in response to pro-inflammatory signals mediated by PKA [32]. The findings of this study, indicating a significant increase in the expression of genes associated with the cAMP pathway, align with this role. It further substantiates that those immune cells respond to heat stress environments by upregulating the expression of genes related to the cAMP signaling pathway.

In this experiment, a further KEGG functional enrichment analysis was conducted on genes exhibiting significant changes. The findings revealed predominant enrichment in pathways associated with protein processing in the endoplasmic reticulum, the MAPK signaling pathway, and the IL-17 signaling pathway (Figure 3). Among the MAPK signaling cascades identified, the JNK and p38 MAPK pathways are primarily linked with cellular stress and apoptosis [33,34]. The p38 MAPK pathway additionally functions as a classical inflammatory pathway [35]. Interestingly, this study observed gene enrichment within the MAPK signaling pathway following heat stress and LPS treatments. This suggests that when cells are exposed to external stress and inflammatory factors, they can activate the MAPK signaling pathway, inducing an inflammatory response and modulating its intensity and duration. IL-17, a pro-inflammatory cytokine, and the interleukin-17 signaling pathway play critical roles in the pathogenesis of related inflammatory diseases [36]. The experiment highlighted a significant upregulation of gene expression associated with the IL-17 signaling pathway. Numerous homologous proteins within the IL-17 family have demonstrated roles in tumor development, cellular inflammation, and similar processes [37]. For instance, IL-17B elevation is linked to intestinal inflammatory responses, promoting neutrophil migration. IL-17D, on the other hand, increases during tumors and viral infections [38], stimulating endothelial cells to trigger classical pro-inflammatory cytokine responses [37]. In summary, the study’s results indicated a substantial enrichment of genes within inflammatory pathways. This suggests that immune cells respond to heat stress and LPS stimulation by inducing inflammatory responses through the MAPK and IL-17 signaling pathways, thereby influencing the duration and intensity of cellular inflammation.

## 5. Conclusions

Heat stress and LPSs individually trigger immune responses, while LPSs intensify the immune response induced by cellular heat stress. When infected in combination, heat stress and LPSs collectively reduce the expression of CRHR1, MEOX1, and MOV10L1. This combination elicits an inflammatory response through pathways involving cAMP, IL-17, MAPK, and other pathways.

## Figures and Tables

**Figure 1 animals-14-00532-f001:**
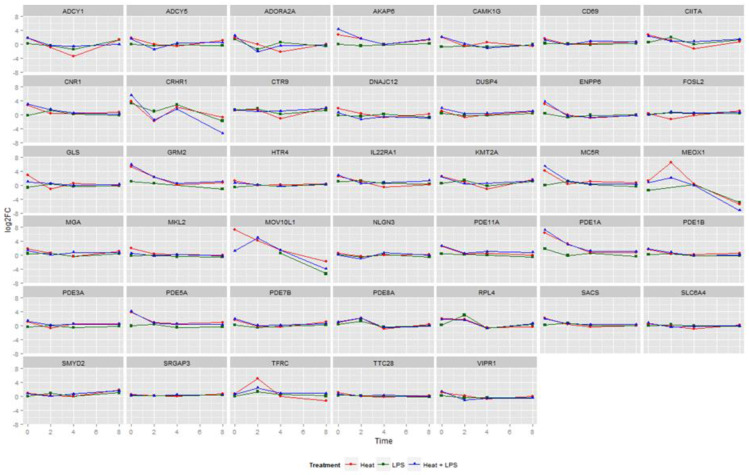
Logarithmic changes in gene expression across treatments. Line chart of multiple ratios of gene expression. The horizontal coordinates are the time points, and the vertical coordinates are the multiples of the expression quantities. The red line is the heat stress treatment group, the green line is the LPS treatment group, and the blue line is the combination of heat stress and LPS treatment group.

**Figure 2 animals-14-00532-f002:**
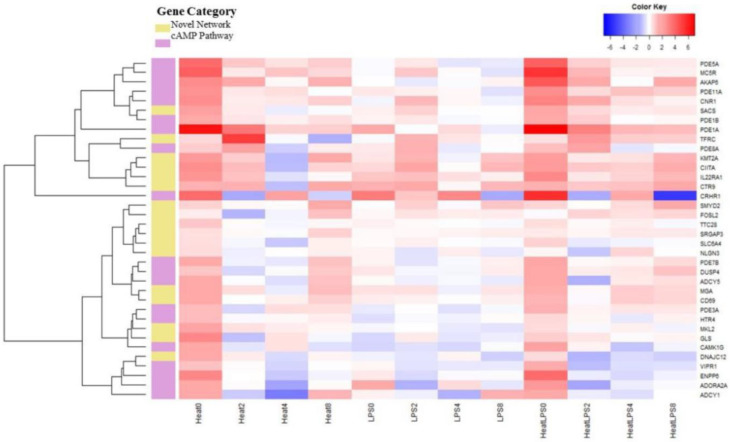
Heat map of gene expression. Heat map of gene pathway enrichment. The ordinate is pathway enrichment and gene name. The horizontal coordinates are the processing groups and processing times. A warm color shows upregulated gene expression, and a cool color shows downregulated gene expression.

**Figure 3 animals-14-00532-f003:**
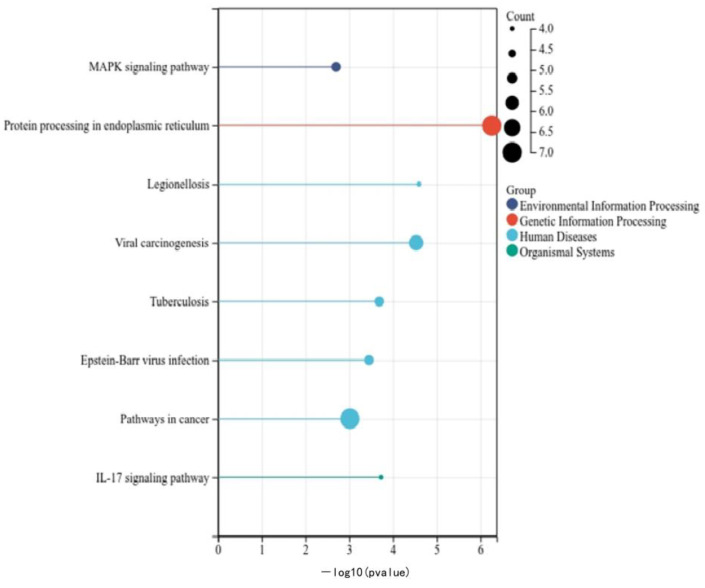
KEGG pathway. KEGG enrichment pathway diagram. The ordinate represents the path name, the abscess represents the P-value. Count: the size of the circle represents the number of enriched genes. The circle color indicates the different functions of the path.

## Data Availability

The original contributions presented in the study are included in the article; further inquiries can be directed to the corresponding author/s.

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
