# Peer review of "Effects of Heat Stress and Lipopolysaccharides on Gene Expression in Chicken Immune Cells"

_animals, 2024, doi:10.3390/ani14040532_

Round 1

Reviewer 1 Report

Comments and Suggestions for Authors

The document is adequately written, no palgiarism or self-citations were detected. I consider it to be a scientifically sound document with content of significance to the scientific community.

Figure 1 may be difficult to read due to the amount of information presented. The title of Figure 2 is not adequately appreciated in the submitted document. The discussion is adequate.

  Some details that need to be clarified in materials and methods: It is not clear the difference between control treatment and blank normal temperature treatment.   Section 2.5 should be revised as it is not clear how mRNA expression was detected in all treatment groups. Based on what criteria were the significance thresholds for the gene sets defined?

In section 2.6 it is mentioned that quality control was performed on the data, but this is not specified.

Author Response

Thank you very much for taking the time to review this manuscript. Your comments will be of great help to the improvement of the writing quality of this paper. 

I have made a comprehensive revision and improvement of this article. Please see the attachment.

Reviewer 2 Report

Comments and Suggestions for Authors

Dears Authors

The research undertaken by the Authors of the manuscript is interesting with significantly aspects of novelty and follows the current research trend in this field. I believe that the experiment was properly carried out using appropriate material and research methods. The obtained results are interesting and valuable. In my opinion, some elements require minor refinement or clarification.

Abstract

Please expand the abbreviations the first time you use them.

Introduction

Lines 40-42 – Please edit the sentence as it can be understood that corticosterone is secreted by the pituitary gland. Corticosterone is secreted by the adrenal cortex; the pituitary gland is the place where its secretion is stimulated.

Please expand the abbreviations the first time you use them.

Material and Methods

Lines 84-85 – The information is repeated in the content of the next subsection.

Line 91 – I understand that the idea was to determine the number of viable cells. Please elaborate on what activity was detected.

Line 114 – Please check the name Fluidigm Corporation. I think it's currently Standard Bio Tools Inc.

Line 121 – Please expand the BMDC abbreviation

Lines 120 and 122 – Please put company names in brackets.

Results

Line 134 – According to Figure 1, a more pronounced decreasing trend was visible from 0 h - to 2 h

Lines 134-135 as well as 169--174 – In my opinion the information should be included in the Discussion section.

Lines 138-139 – Please verify the term "pre-treatment". The results demonstrated in Figure 1 refer to the after-treatment period.

Discussion

Please expand the abbreviations the first time you use them.

Sincerely

Author Response

(The authors gave the same response as above.)

Reviewer 3 Report

Comments and Suggestions for Authors

In this manuscript, the authors quantify the effects of heat and Lipopolysaccharide (LPS) on marker gene expression in poultry immune cells. The topic is interesting and relevant. However, the following issues stand out.

Firstly, the experimental procedure and methods are not well-documented. The catalog numbers for all reagents and materials should be provided.

Additionally, the sentence “Interactions between strains and treatments were fitted to the ANOVA model.” in line #125 is problematic, as there is no "strains" variable in this study.

The authors selected 48 candidate genes based on a previous study, as stated: “Initial experiments conducted a pre-transcriptome sequencing analysis using IPA, identifying 48 interacting genes potentially associated with heat stress and LPS [20].” However, the referenced literature pertains to "avian leukosis virus," not heat stress and LPS. This is not a trivial mistake.

The description of the results is poor and needs to be rephrased and improved. The current description reads more like pieces of discrete figure legends. There is no coherent organization.

The results presented in Figure 1 should be supported by statistical analysis. For instance, the significance (p-value) of the time series, treatment, or their interaction should be provided.

Regarding the results of Figure 3, the analysis procedure or code should be provided. Specifically, my concern is that the authors' use of 48 candidate genes introduces a bias towards the KEGG pathway from the outset. The background should be set against the 48 candidate genes, not the entire KEGG gene sets. Thus, the results of Figure 3 should be carefully evaluated.

Comments on the Quality of English Language

The language of this manuscript requires improvement. For example, the opening sentence, “Poultry are thermoregulatory creatures,” is problematic.

Author Response

Thank you very much for taking the time to review this manuscript. Your comments will be of great help to improve the quality of this article.

I have made a comprehensive revision and improvement to this article. Please refer to the attachment.

Round 2

Reviewer 3 Report

Comments and Suggestions for Authors

Thanks to the author's efforts, the revised manuscript shows clear improvements. However, the description of the KEGG enrichment analysis remains vague and would benefit from further elaboration. Specifically, the statement, "Based on the enrichment analysis of 48 different expression genes, a total of 134 channels were enriched, including 18 significantly enriched pathways," is unclear. The term "different expression gene" is ambiguous; it would be helpful to clarify whether this refers to "differentially expressed genes." Additionally, it is not evident whether all 48 genes are differentially expressed since only 48 candidate genes exist. The criteria or methodology used by the author to identify and define these 48 genes as "different expression genes" are not explicitly stated, and providing this information would enhance the clarity and rigor of the analysis.

Comments on the Quality of English Language

Minor editing of the English language required

Author Response

Thank you very much for taking the time to review this manuscript. Your comments will be of great help to the improvement of the writing quality of this paper.
